# Key Sweet Potato Viruses in Fujian Province and Their Distribution, Harmfulness, and Implications in China

**DOI:** 10.3390/cimb47040242

**Published:** 2025-04-01

**Authors:** Weikun Zou, Shi-Peng Chen, Zhijian Yang, Xuanyang Chen

**Affiliations:** 1College of Agronomy, Fujian Agriculture and Forestry University, Fuzhou 350002, Chinayangzj41@163.com (Z.Y.); 2Department of Horticulture, National Chiayi University, Chiayi 600355, Taiwan; tw00058329@gmail.com; 3Cross-Strait Agricultural Technology Cooperation Center under the Ministry of Agriculture and Rural Affairs, Fuzhou 350002, China

**Keywords:** sweet potato, virus disease, SPVD, *RNase3*, Fujian province

## Abstract

China, the largest global producer of sweet potatoes, faces significant threats from viral diseases, particularly in Fujian Province, where sweet potatoes are the second most important food crop after rice. This study identified 11 viruses, including *sweet potato feathery mottle virus* (SPFMV) and *sweet potato chlorotic stunt virus* (SPCSV), infecting sweet potatoes in Fujian. Sequence comparisons revealed diverse strains from various sources. Virus prevalence varied across regions, with Quanzhou, Fuzhou, and Putian severely affected, detecting 10, 9, and 7 viruses, respectively, compared to only 3 in Sanming and Longyan. In particular, sweet potato virus disease (SPVD) caused the most severe damage during the seeding stages, resulting in dwarfing and leaf deformation, while the damage was lighter during the growth period, manifesting as the yellowing and brittleness of the leaves, ultimately reducing the yield. Compound infestations predominated, with between 0 and 6 viruses infecting different sweet potato varieties. Single-virus infections were observed for *sweet potato virus 2* (SPV2), *sweet potato symptomless virus 1* (SPSMV-1), and *sweet potato pakakuy virus* (SPPV), while others, particularly SPCSV, were frequently co-infected with SPFMV, leading to SPVD development. Further analysis showed that the *RNase3* expression of SPCSV was correlated with the SPVD severity in sweet potato. These findings provide insights into the epidemiology of sweet potato viruses and serve as a reference for developing targeted disease management strategies.

## 1. Introduction

Sweet potatoes, introduced to China through Fujian Province, play a crucial role in food security and are the second most cultivated food crop in Fujian after rice. However, sweet potato production is threatened by viral diseases. More than 30 virus species, spanning families such as *Bromoviridae*, *Bunyaviridae*, *Caulimoviridae*, *Closteroviridae*, *Comoviridae*, *Flexiviridae*, *Geminiviridae*, *Luteoviridae*, and *Potyviridae*, have been reported to infect sweet potatoes, including RNA viruses like sweet potato chlorotic stunt virus (SPCSV), sweet potato feathery mottle virus (SPFMV), and SPMSV (sweet potato mild speckling virus), as well as DNA viruses like sweet potato leaf curl virus (SPLCVS), sweet potato symptomless virus 1 (SPSMV-1), and sweet potato pakakuy virus (SPPVS) [1,2,3,4,5,6,7]. Eight of these virus species have been identified in China, namely SPFMV, SPVG, SPLV, SPCFV, SPCSV, SPVMV, CMV, and SPLCV, among which the first six exist in Fujian Province, with SPFMV being the most prevalent [8].

A number of factors drive the high incidence (60–90%) of viral diseases in sweet potato production, such as vector populations (aphids and whiteflies), crop resistance, soil conditions, and cropping systems [9,10,11]. Viral diseases significantly reduce sweet potato yield and quality [12]. Sweet potatoes reproduce asexually, allowing viruses to accumulate and transmit across generations [13]. Infected plants exhibit reduced tuber size, cracked skin, and yield losses [9,10,11,12,13,14]. Compound viral infestations enhance vector transmission efficiency, expand host ranges, and increase the number of susceptible varieties [15,16,17].

Sweet potato virus disease (SPVD) is caused by the co-infection of two such strains, the aphid-transmitted SPFMV and whitefly-transmitted SPCSV, posing a significant threat with up to an 80% yield loss in high-production areas. Initially identified in Africa in the 1970s, the disease was first reported in China in 2012 and is now prevalent in key sweet potato-producing provinces, including Fujian [18,19,20,21,22]. Different strains have been found within most virus species. For example, SPFMV can be categorized into three strains: the common type (SPFMV-O), the brown crapevirus type (SPFMV-RC), and the East African type (SPFMV-EA) [18], while SPCSV can be categorized into two strains, namely the East African type (SPCSV-EA) and the West African type (SPCSV-WA) [23,24,25].

Due to the severe damage that SPVD can cause, its underlying mechanisms have become an important focus of research. Studies show that the *RNase3* gene of SPCSV plays a critical role in enhancing SPVD severity by suppressing sweet potato gene silencing, thus facilitating viral invasion [26,27,28,29]. However, the potential application of *RNase3* in controlling sweet potato viral diseases remains underexplored. Further exploration of *RNase3* could provide valuable insights for managing SPVD and other viral diseases in sweet potatoes.

In this study, eleven major virus diseases of sweet potatoes in the main production areas of Fujian Province were identified. The differences in virus disease infections among different varieties in different regions were investigated, as well as the differences in damage caused by SPVD at different development stages of sweet potatoes. Additionally, the role of the *RNase3* gene in SPVD severity was preliminarily analyzed. The results provide a scientific basis for the prevention and control of virus diseases in sweet potatoes, thereby ensuring production safety, and also serve as a reference for the development of anti-virus breeding in sweet potatoes.

## 2. Materials and Methods

### 2.1. Sweet Potato Sampling

Sixty-three sweet potato samples (41 varieties, including widely planted varieties such as Jin57, Jin3, Quan12, Funingzi3hao, Shenglibaihao, and Fu18) with suspected virus symptoms were collected from Fuzhou, Quanzhou, Putian, Sanming, and Longyan. One or more leaves or tuberous roots were collected according to the symptom of the virus disease. The sample collection locations are shown in Figure 1. These selected locations are important sweet potato production areas in Fujian Province, and the sampled varieties are mostly planted in Fujian Province. The DNA and RNA were extracted immediately post-collection for analysis.

### 2.2. Reagents

Key reagents included the RNAperp Pure Polysaccharide Polyphenol Plant Total RNA Extraction Kit; the plant genome DNA extraction kit; the universal DNA purification and recovery kit; the common plasmid small extraction kit, which included ethidium bromide (EB), dNTPs, and DNase I (RNase-free) (TIANGEN); and the reverse transcription kit, which included Taq DNA polymerase, 10× reaction buffer, the 25 mM magnesium chloride solution, a 200 bp DNA ladder, and agarose (Promega, Madison, WI, USA). Ethanol, sucrose, and other commonly used reagents were locally sourced in China.

### 2.3. PCR Analysis

#### 2.3.1. Primer Design

According to GenBank sequences, primers targeting conserved the *cp* (coat protein), *HSP70*, and *RNase 3* genes were designed with Primer 5.0 and DNAMAN software (7.0) or sourced from the literature [8,27,30,31,32,33]. Primer synthesis was conducted by Invitrogen (Shanghai, China). Stable and specific primers were selected for large-scale field sample detection (Appendix A).

#### 2.3.2. RNA and DNA Extraction

RNA and DNA were extracted from infected leaves and tuberous root skin using the RNAperp and TIANGEN kits, respectively. Leaves with no disease (4–6 leaves, ~0.1 g) were sampled with a 1 cm diameter puncher, while the root skin (~0.1 g) was sliced with a blade.

The leaf samples were grinded in the tubes with a sterilized small steel ball, and the roots were grinded in sterilized mortars. RNA quality (A_260/280_ ≈ 2.0, A_260/230_ = 1.8–2.2) and DNA quality (A_260/280_ = 1.8; A_260/230_ = 1.8–2.2) were analyzed using a Nanodrop 2000 ultra microspectrophotometer and further validated via 1.0% agarose gel electrophoresis. RNA and DNA were stored at −80 °C and −20 °C, respectively.

#### 2.3.3. PCR and RT-PCR Protocols

For reverse transcription, 4 µL of RNA and 1 µL of Oligo (dT) 15Primer (500 µg/mL) were combined at 70 °C for 5 min. The mixture was then placed on ice for 5 min. After that, 1 µL of the GoScript™ reverse transcriptase, 4 µL of the GoScript™ 5 × Reaction Buffer, 1.5 µL of MgCl_2_ (25 mM), 0.5 µL of the Recombinant RNasin^®^ Ribonuclease Inhibitor, 1 µL of the PCR Nucleotide Mix (10 mM), and 7 µL of nuclease-free water were added, followed by incubation at 25 °C for 5 min, 42 °C for 80 min, and 70 °C for 15 min.

PCR was performed in a 25 µL reaction system consisting of the cDNA template, 10× PCR buffer, dNTP, MgCl_2_, Taq polymerase, gene-specific primers, and ddH_2_O. When one variable was optimized, the other reaction conditions remained constant. The following parameters were optimized for template dilutions (0×, 10×, 20×, 50×, 100×, and 1000×), dNTP concentrations (1.0, 1.5, 2.0, and 2.5), MgCl_2_ concentrations (1.0, 1.5, 2.0, and 2.5), Taq polymerase concentrations (0.1, 0.2, and 0.3), primers concentrations (0.1, 0.2, 0.5, and 1.0), annealing temperatures (52 °C, 54 °C, 56 °C, 58 °C, 60 °C, 62 °C, 64 °C, 66 °C, 68 °C, and 70 °C), and cycle numbers (30 and 35 cycles). The PCR products were then analyzed using 1.0% agarose gel electrophoresis.

### 2.4. Identification of PCR Products

Specific PCR fragments were excised from 1.0% agarose gel and then purified using the TIANGEN universal DNA purification recovery kit. The purified fragments were cloned into a T-vector, and the sequences were confirmed through DNA sequencing (Invitrogen).

### 2.5. Sequence Analysis

The gene sequences were aligned with the NCBI database using BLAST (https://blast.ncbi.nlm.nih.gov/Blast.cgi, accessed on 26 March 2025). Gene sequences from closed related viruses were downloaded for further phylogenetic analysis, which was conducted using the DNAMAN software and MEGA 7.0.

### 2.6. qRT-PCR to Detect RNase3 Expression

Leaves exhibiting varying degrees of SPVD symptoms were quickly frozen in liquid nitrogen and stored at −80 °C.

Primers for *RNase3* qRT-PCR were as follows (expected product size: 96 bp):

Forward: 5′-GCGAAAGCTATGGTGGAGTCAA-3′;

Reverse: 5′-CCCACCACCGAAAGTCATTCTA-3′.

Primers for the internal reference gene, *GAP* (Glyceraldehyde-3-phosphate dehydrogenase, GAP), were as follows [34] (expected size: 91 bp):

Forward: 5′-GCAGGAACCCGGAAGAGATT-3′;

Reverse: 5′-CAGCCTTGTCCTTGTCAGTG-3′.

### 2.7. qRT-PCR Conditions

The qRT-PCR analysis was performed using the Applied Biosystems StepOne™Real-Time PCR System. The reaction volume (10 µL) consisted of the following: 2 µL of cDNA, 5 µL of the 2 × GoTaq^®^ qPCR Master Mix, 0.1 µL of the forward primer, 0.1 µL of the reverse primer, 2.7 µL of nuclease-free water, and 0.1 µL of the CXR reference fluorescent dye. The qRT-PCR preparations consisted of one cycle at 95 °C for 2 min, followed by 40 cycles at 95 °C for 15 s, 55 °C for 30 s, and 72 °C for 30 s, using the Applied Biosystems StepOne™Real-Time PCR System (Applied Biosystems Inc., Foster City, CA, USA). The melting curve was then generated.

Three replicates were conducted for each experiment. Data analysis was performed using the StepOne Software v2.3, and *RNase3* gene expression was calculated using the 2^−ΔΔCt^ method.

## 3. Results

### 3.1. Detection of 11 Viruses in Fujian Sweet Potatoes

Sweet potatoes in Fujian Province were found to host multiple viruses, leading to complex infections and various typical symptoms (Figure 2). Viruses tend to accumulate in sweet potato plants over successive generations, compounding the infections. The RT-PCR analysis using virus-specific primers (Appendix A) identified 11 viruses infecting sweet potatoes in the region. Key findings for each virus are as follows: 

#### 3.1.1. SPFMV (Sweet Potato Feathery Mottle Virus)

Among 24 SPFMV isolates from sweet potato varieties such as Jin57, Guang79, and others, most belonged to the EA strain (Appendix A). Homologous pairs were identified in Guang79, Jin3, 13286, and Zhan271, as well as in Nongda1 and Longjin1. Some isolates (e.g., those from Puhang1, Quan32, and Fu18) exhibited similarity to both the EA and O strains, whereas no RC strain was detected.

#### 3.1.2. SPCSV (Sweet Potato Chlorotic Stunt Virus)

Four SPCSV isolates were detected in the varieties 13286, Maishu8, Quan32, and Jin3 (Appendix A). Isolates from 13,286 and Maishu8 were closely related to Spanish Can181-9, while those from Quan32 and Jin3 were closely related to the Anhui-11-2 strain from Anhui Province, China. Only West African (WA) strains were observed, with no evidence of EA strains.

#### 3.1.3. SPV2 (Sweet Potato Virus 2)

Three SPV2 isolates were obtained from Mianshu8, Jin57, and Nongda1. Isolates from Mianshu8 and Jin57 were closely related and clustered with isolates from the United States (GWB-2) and Spain (AM-MB2). The isolate from Nongda1 was closely related to the Korean isolates SCN20 and GJ118. These results indicate diverse infection sources (Appendix A).

#### 3.1.4. SPVC (Sweet Potato Virus C)

Eight SPVC isolates were obtained from the varieties Jin15, Zhan271, Jjc4, 13286, Jin57, Longjin1, Guang08-6, and Guang79. These isolates formed three distinct groups (Appendix A).

Jin15 and Zhan271 grouped with previously reported isolates such as 51-9S, Spain1C, SPVC AM-MB2, HN52, and SPVC-ARG.

Jjc4, 13286, Jin57, and Longjin1 grouped with 19-T, Bungo, CW135, 3817-2, and UN202.

Guang08-6 and Guang79 grouped with the Israeli IL isolate. This suggests multiple infection sources or a higher mutation rate in SPVC.

#### 3.1.5. SPVG (Sweet Potato Virus G)

Five SPVG isolates were isolated from Quan32, Quan12, Jin57, Longjin1, and Guang79 and formed three groups (Appendix A):

Quan32 grouped with SC6.

Jin57 and Longjin1 were closely related, forming a separate group.

Quan12 and Guang79 grouped with previously reported isolates including SPVg-Jianshui8, Belgium, Shandong7, Shaanxi4, and Guangdong7.

These findings suggest diverse infection origins or a high mutation potential in SPVG.

#### 3.1.6. SPLV (Sweet Potato Latent Virus)

A single SPLV isolate was detected across different varieties. Sequence analysis showed close homology to other SPLV isolates previously reported in China (Appendix A).

#### 3.1.7. CMV (Cucumber Mosaic Virus)

A single CMV isolate was obtained from the variety Quan12, showing the closest relationship to the DZ-2 isolate from Guangdong Province. This suggests that the infection source may have originated from Guangdong (Appendix A).

#### 3.1.8. SPLCV (Sweet Potato Leaf Curl Virus)

One SPLCV isolate was identified in the variety Jin15, exhibiting the closest relationship to the SPLCV-MEX isolate from Mexico (Appendix A). This suggests that SPLCV may be relatively stable and less prone to mutation.

SPPV (Sweet Potato Pakakuy Virus) and SPSMV-1 (Sweet Potato Symptomless Virus-1).

SPPV and SPSMV-1 isolates were detected in Fuzhou4 and Quan32, respectively. Both isolates were closely related to Huachano1 isolates from Peru (Appendix A).

#### 3.1.9. SPVMV (Sweet Potato Vein Mosaic Virus)

An SPVMV isolate was identified in Mianshu8, with the closest relationship to the Henan isolate from China (Appendix A).

### 3.2. Regional Variations in Sweet Potato Virus Infections in Fujian Province

Sweet potato virus diseases varied across regions in Fujian Province, with differences in virus types and incidence rates (Table 1). The most prevalent virus was potato Y virus, accounting for 53% of incidences, followed by maize lineovirus (17%) and baculovirus (16%). In contrast, CMV and bean golden yellow mosaic virus were the least common, at 4% and 1%, respectively (Figure 3).

SPV2 and SPPV were the most widespread and were detected in all experimental areas. SPSMV-1 was present in all regions except Putian, while SPVMV and CMV were confined to Quanzhou.

### 3.3. Variations in Sweet Potato Virus Incidence Across Growth Stages

The incidence of viral diseases varied across the different growth stages of sweet potatoes (Table 2). At the seedling stage, all 11 viruses—SPFMV, SPCSV, SPVC, SPVG, SPV2, SPLV, SPVMV, CMV, SPLCV, SPPV, and SPSMV-1—were detected, with SPLV and CMV found exclusively at this stage. This suggests that the seedling stage is optimal for comprehensive virus investigation.

In the early field growth stage, SPCSV, SPVC, and SPV2 had the highest detection rates, at 50%, 44%, and 88%, respectively, indicating that their activity peaks during this stage. SPVMV, SPLCV, and SPSMV-1 were most frequently detected, with respective rates of 56%, 11%, and 75%, In the late growth stage, SPFMV and SPPV showed the highest detection rates, at 100% and 70%, respectively.

Detection rates for individual viruses varied across stages. For instance, SPVMV was rarely detected during early-stage field growth. The same was observed for SPCSV during the prime growth stage and SPV2 during the late growth stage, with a detection rate of 0%. This variation may be due to virus suppression at specific stages.

Notably, SPCSV and SPVC exhibited similar trends across growth stages, suggesting potential synergistic interactions between the two viruses.

### 3.4. SPVD Detection Across Sweet Potato Growth Stages

SPVD detection rates varied significantly across growth stages, with 27%, 64%, 0%, and 15% detected at the seedling stage, early field growth stage, peak growth stage, and late growth stage, respectively (Table 3). This indicates that SPVD is most severe during early field growth but is only minimally detectable during the peak growth stage. This phenomenon, referred to as “high temperature cryptosis”, suggests that SPVD may be inhibited by elevated temperatures during peak growth. The trend in SPVD detection closely mirrored that of SPCSV, indicating that SPVD occurrence is mainly influenced by SPCSV.

### 3.5. Virus Variations Among Sweet Potato Varieties

Different sweet potato varieties exhibited distinct patterns of viral infections (Table 4). Among the 41 tested varieties, 39 (95.12%) showed viral infections, with some experiencing co-infections by multiple viruses, including up to six viruses detected simultaneously.

Single-virus infections were detected in 6 varieties, two-virus infections in 6 varieties, three-virus infections in 8 varieties, four-virus infections in 4 varieties, five-virus infections in 11 varieties, and six-virus infections in 4 varieties. Virus types also varied. DNA viruses were only found in varieties such as Puzi5hao, Shenglibaihao, SM2, Jin11, SM3, FZ1, Rong910, and 11-500; RNA viruses were only detected in Longjinshu1hao, Pu3hong, Mianshu8, Fu18, and QZJJC4; mixed RNA and DNA viruses were detected in 28 varieties. These results highlight the diversity of viral infections among sweet potato varieties in Fujian Province.

### 3.6. Occurrence of Viral Diseases in Different Regions

In coastal areas, sweet potato varieties from Putian, Fuzhou, and Quanzhou are more frequently exchanged with the outside world, resulting in the highest detection rate of viral diseases in samples, especially in Putian, where it reaches 50% (Table 5). In contrast, the detection rates in inland areas such as Sanming and Longyan are 0%, as their sweet potato seedlings are relatively less exchanged with the outside areas. This highlights the importance of quarantine measures during seedling transportation for the prevention and control of viral diseases.

### 3.7. Complex Infections of Sweet Potato Viral Diseases

Among the 41 sweet potato varieties tested in this study, 33 (80.49%) were infected by two or more viruses. The co-infection rate for RNA viruses was 51.22%, while the co-infection rate for DNA viruses was 53.66%. Key patterns of co-infection include the following:

Two-virus co-infections:

SPPV + SPSMV-1: 7.31%;

SPV2 + SPSMV-1: 4.87%;

SPFMV + SPSMV-1: 2.43%.

Three-virus co-infections:

SPV2 + SPPV + SPSMV-1: 7.31%;

SPFMV + SPPV + SPSMV-1: 4.87%;

Other combinations: 2.43%.

Four-virus co-infections:

SPFMV + SPVC + SPPV + SPSMV-1: 4.87%;

Other combinations: 2.43%.

Five-virus co-infections:

SPFMV + SPV2 + SPPV + SPSMV-1 + SPLCV: 7.31%;

Other combinations: 2.43%.

Six-virus co-infections:

SPFMV + SPCSV + SPVG + CMV + SPPV+ SPSMV-1: 2.43%;

Other combinations: 2.43%.

Notably, SPV2, SPSMV-1, and SPPV exhibited single-virus infections with rates of 5.26%, 7.14%, and 11.54%, respectively. Complex infections predominantly involved SPCSV, which was most frequently co-infected with SPFMV. SPFMV was most frequently co-infected with SPSMV-1. Additionally, SPVC, SPFMV, SPV2, SPSMV-1, SPPV, SPCSV, SPLCV, SPVG, SPLV, and SPVMV displayed co-infection patterns, with varying tendencies to synergize with other viruses.

### 3.8. SPVD Detection in Sweet Potato Varieties

Among the tested varieties, nine were co-infected with SPCSV and SPFMV, resulting in an SPVD infection rate of 21.95%. Thirteen varieties (31.71%) were infected with SPFMV alone, which could lead to SPVD when co-infection with SPCSV occurs. Only one variety, Jin15, was solely infected with SPCSV (2.3%).

### 3.9. Occurrence of Viral Diseases in Different Sweet Potato Parts

The occurrence of viruses varied between sweet potato leaves and tubers (Table 6). The detection results for other viruses, including SPCSV, SPVC, SPVG, CMV, SPPV, and SPSMV-1, differed between plant parts. For example, SPVG was detected in the leaves of Guang79 but not in tubers. SPVC was not detected in the leaves of Fu18 and Zhan271 but was found in their tubers. SPCSV was more readily detected in the leaves of Quan12 but predominantly appeared in the tubers of Fu18. SPVD was detected in the leaves of Quan12 and the potato blocks of Fu18. These results suggest that viruses accumulate at varying concentrations in different plant parts depending on the time and type of virus.

### 3.10. Correlation Between RNase3 Gene Expression and SPVD Pathogenesis

The relative expression levels of RNase3 in different samples are shown in Figure 4. From highest to lowest expression, the samples are ranked as follows: Jin 3 (May), Jin 3 (July), Mianshu 8, Jin 3–82 (green leaf), Jin 3–82 (yellow leaf), Guang 87–72 (July), and Guang 87–72 (April).

There were many trends in symptom severity. Among these, Jin 3 (May) exhibited the most serious symptoms, including bright vein pulses, chlorosis, distortion, and deformity. Jin 3 had relatively mild symptoms in July and May, and no clear pulse appeared. Mianshu 8, Jin 3–82, and Guang 87–72 all displayed deformity symptoms, with Mianshu8 showing the most severe symptoms and Guang 87–72 displaying the mildest.

Samples exhibiting bright vein pulse symptoms showed higher *RNase3* expression compared to those with deformity symptoms. Elevated RNase3 expression was directly associated with more severe SPVD symptoms. The expression levels of *RNase3* varied over time and between plant parts, with its expression consistently higher in SPVD samples than in those infected solely with SPCSV.

## 4. Discussion

### 4.1. Viral Diseases Threaten Sweet Potato Production

Our research results indicate that sweet potato viruses have been detected in most parts of Fujian Province, and the most severe SPVD has also been reported in many regions across China. Therefore, virus diseases have become a major threat to sweet potato production and require significant attention. In fact, sweet potato virus disease is highly prevalent in agricultural production. Statistics show that the incidence rate can reach 60 to 90%, making it one of the key limiting factors in sweet potato production [35]. In addition, some sweet potato viruses show ‘high temperature hidden symptoms’ during high-temperature seasons. This suggests that the infection rate of sweet potato virus disease may be much higher than observed.

Virus infection is a major factor contributing to the reduction in sweet potato yield and the degeneration of sweet potato varieties. Since sweet potatoes reproduce asexually, viruses can easily accumulate. Once infected, the viruses are transmitted from generation to generation. Infected plants may exhibit weak seedling growth and slow regeneration.

Infected plants exhibit reduced yield components, including fewer and smaller tubers, rough skin texture, and brown cracked tubers. In the Americas and Africa, virus diseases have caused a 20% to 57% reduction in sweet potato yields [36,37]. Sweet potato virus disease (SPVD), resulting from the synergistic infection of SPFMV and SPCSV, has a significant impact on yield. In severe cases, it can cause a yield loss exceeding 90%, or even total crop failure. In China, the economic losses caused by sweet potato virus diseases are estimated to reach CNY 4 billion annually [38].

### 4.2. Symptoms of Sweet Potato Viral Diseases

The main symptoms of sweet potato virus infection are as follows: (1) Leaf spotting: This can occur in both the seedling and field stages. In the early stage of the disease, the leaves have clear veins and may also have chlorotic translucent spots. As the disease progresses, affected areas turn purple-brown, forming purple spots, purple ring spots, yellow spots, or necrotic lesions. Most varieties develop characteristic purple feather spots along the veins, while a few varieties exhibit only chlorotic transparent spots. (2) Mosaic leaf type: Infected seedlings initially show transparent veins arranged in a reticular pattern. Over time, irregular yellow-green mosaic patterns appear along the veins. (3) Curled leaves: the edges of the leaves curl upward and may form a cup shape in severe cases. (4) Wrinkled leaves: Diseased seedlings develop small, wrinkled leaves with uneven or twisted edges. The chlorotic translucent spots may also appear parallel to the midrib. (5) Yellow leaves: this includes leaf yellowing and the presence of reticular yellow veins. (6) Cracked tubers: Dark brown or yellow brown cracks develop on the tubers. (7) Arbuscule type: the whole plant is clustered, and the leaves are obviously small.

A serious SPVD infection in sweet potatoes causes leaf twisting, deformation, chlorosis, prominent veins, and severe dwarfing. An infection at the seedling stage may result in total crop failure, while an infection at the later stage of production may result in a significant reduction in yield and commodity quality. SPCSV, one of the viruses responsible for SPVD, belongs to the genus *Trichovirus* and has the ability to synergize with various viruses. When SPFMV and SPCSV infect sweet potatoes individually, the sweet potatoes show no or only mild symptoms. However, co-infection with both viruses leads to severe symptoms. In addition, the SPFMV levels increase by 600 times in co-infected plants compared with those infected with SPFMV alone, whereas the SPCSV levels remain unchanged or decrease slightly.

### 4.3. Prevention and Control of Sweet Potato Viral Diseases

It is speculated that SPCSV may inhibit or interfere with the antiviral defense pathways of sweet potato plants, thereby causing their resistance to SPFMV infection and leading to increased viral accumulation [27,28,39]. This suggests that the damage caused by co-infection is significantly more severe than that caused by a single infection.

Once sweet potatoes are infected with a virus, eliminating it with pesticides is difficult. The simplest and most effective way for preventing and controlling viral diseases in production is the use of virus-free seedlings. However, the high cost of virus-free seedling limits their widespread application. Therefore, additional preventive and control measures can be implemented to ensure the safety of sweet potato production.

First, strengthen quarantine measures for sweet potato virus diseases. The long-distance transmission of sweet potato viruses is mainly carried by seed tubers and seedlings. Cross-infection can happen during germplasm exchange of sweet potatoes and the circulation of seedlings. Therefore, rigorous virus screening is essential when introducing new sweet potato varieties. Once a harmful virus is detected, the infected seed tubers and seedlings should be destroyed in time to prevent the spread of viral diseases.

Second, improve the accuracy in viral disease detection. Accurate virus detection is the basis for effective prevention and control. Currently, three main detection methods exist. The first is the indicator plant grafting method, which uses Brazilian morning glory as an indicator plant. However, this method is slow and time-consuming. The second is the serological method, specifically the ELISA method, which is costly and requires improved accuracy. The third is the PCR method, which, although highly sensitive, requires laboratory conditions, is expensive, and also needs accuracy enhancements. Therefore, further research and development efforts are needed to establish faster and more user-friendly detection methods. For example, a convenient nucleic acid detection system can be developed based on the rapid and sensitive CRISPR technology and AI for the detection of sweet potato virus diseases.

Third, strengthen field management of sweet potato production. Sweet potato virus disease can affect sweet potatoes from the seedling stage to harvest, with both seedlings and tubers susceptible to infections. The virus can be spread through grafting juice and insect feeding. Therefore, effective field management throughout the growth period is essential to reduce the virus rate in seed tubers and seedlings, as well as to minimize the population of virus vectors, thus preventing the spread of viral diseases. To strengthen field management, it is necessary to promptly remove diseased sweet potato plants while avoiding mechanical damage to healthy plants and preventing contact with viral sap during the removal process. Diseased seedlings should be kept as far away from healthy ones as possible, and it is best to bury and destroy them. Additionally, effective pest control, particularly targeting aphids and whiteflies, key insect vectors of virus transmission, should be implemented.

Fourth, focus on sweet potato virus disease resistance breeding. This study shows the susceptibility of different sweet potato varieties to virus diseases, indicating that some varieties exhibit higher resistance to viral diseases. It is possible to breed new varieties that are relatively resistant to viruses by screening more resistant or virus-tolerant varieties for hybridization. In addition, with the application of biotechnology, genetic engineering could offer a viable future approach, allowing for the transfer of anti-viral genes into sweet potatoes to create transgenic plants resistant to viral diseases. Currently, there have been some successful reports of sweet potatoes gaining virus resistance through genetic modification, but practical applications remain distant [26]. In the future, increased investment in scientific research will be essential to achieve new breakthrough results.

Over 30 viruses are known to infect sweet potatoes [40]. In Fujian Province, this study identified 11 virus types infecting sweet potatoes. Most SPFMV strains were classified as the EA strain, while SPCSV strains were predominantly of the WA strain. SPV2, SPVC, and SPVG exhibited significant strain variability, while SPLV, CMV, SPLCV, SPVMV, SPPV, and SPMSV-1 were highly homologous to strains reported in China and internationally, indicating stable infection sources. Regional variations in virus incidence are likely attributed to differences in germplasm exchange frequencies.

This experiment’s reliability is supported by its design, although the sample size may have influenced the detection results. Virus detection could be further affected by hyperthermia [41], as infected plants may exhibit no symptoms at certain temperatures. This may explain the discrepancies, such as the absence of SPVD detection in the Longyan area, in contrast to a previous report [42]. Furthermore, the high detection rates in this study reflect targeted sampling of symptomatic plants and may not directly correlate with field incidence, but they do indicate a consistent trend.

Virus detection was highest during the seedling stage, suggesting that virus accumulation varies across growth stages. This highlights the importance of conducting virus detection during optimal periods to ensure accuracy. Additionally, the distribution of viruses varied across different parts of sweet potato plants. For example, SPCSV was more readily detected in leaves of some varieties and in potato blocks of others. Testing multiple plant parts enhances detection accuracy.

Composite infections were observed in 80.49% of the varieties, with up to six viruses detected simultaneously. SPCSV and SPFMV frequently co-infected sweet potatoes, causing severe SPVD. Synergistic effects between SPCSV and other viruses were evident, but interactions involving other viruses require further exploration.

### 4.4. The Relationship Between the Expression of the RNase3 Gene and SPVD

The expression of the *RNase3* gene is correlated with SPVD severity in infected plants. Samples exhibiting symptoms, such as bright pulses and chlorosis, showed higher *RNase3* expression than those with distortion or vascular abnormalities. *RNase3* expression was higher in SPVD-infected Guang 87-72 plants compared to those infected only with SPCSV, indicating that SPFMV may enhance SPCSV replication. The differences in *RNase3* gene expression between green and yellow leaves in Jin3-82 indicate that enzyme activity regulating *RNase3* expression varies by developmental stage and organ, with higher activity observed in green leaves [29,43,44].

## 5. Conclusions

A total of 11 sweet potato viruses have been identified in Fujian Province, though the extent of damage varies across different regions. Sweet potato virus diseases, particularly SPVD, are spreading at an increasing rate, emphasizing the urgent need for effective prevention and control strategies. These findings highlight the complexity of SPVD pathogenesis and the role of the *RNase3* gene. Further research is necessary to silence the *RNase3* gene via CRISPR-based approaches to determine whether altering *RNase3* expression can mitigate SPVD symptoms or inhibit its occurrence, providing potential strategies for the prevention and control of SPVD.

## Figures and Tables

**Figure 1 cimb-47-00242-f001:**
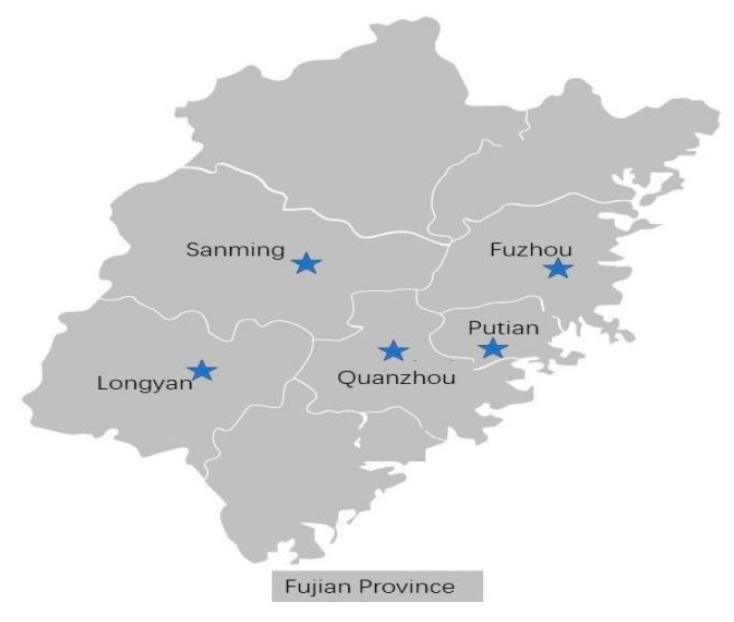
The location of sample collection.

**Figure 2 cimb-47-00242-f002:**
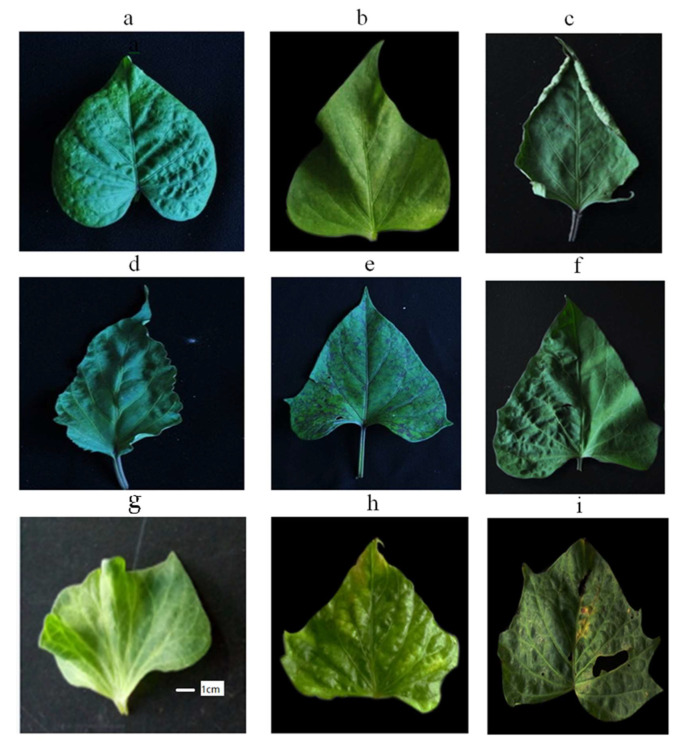
Typical symptoms of virus-infected sweet potato leaves. (**a**) SPPV was detected. (**b**) SPCSV and SPSMV-1 were detected. (**c**) SPLCV and SPFMV were detected. (**d**) SPV2, SPPV, and SPSMV-1 were detected. (**e**) SPFMV, SPVC, and SPPV were detected. (**f**) SPVC, SPVG, and SPFMV were detected. (**g**) SPVD, SPV2, and SPVG were detected. (**h**) SPVMV, SPVD, SPV2, and SPLV were detected. (**i**) SPLV, SPFMV, SPVC, SPVG, SPV2, and SPSMV-1 were detected.

**Figure 3 cimb-47-00242-f003:**
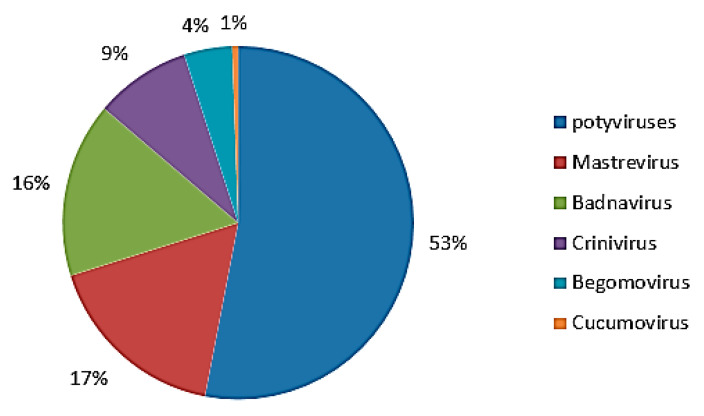
Proportional distribution of sweet potato virus genera in Fujian Province.

**Figure 4 cimb-47-00242-f004:**
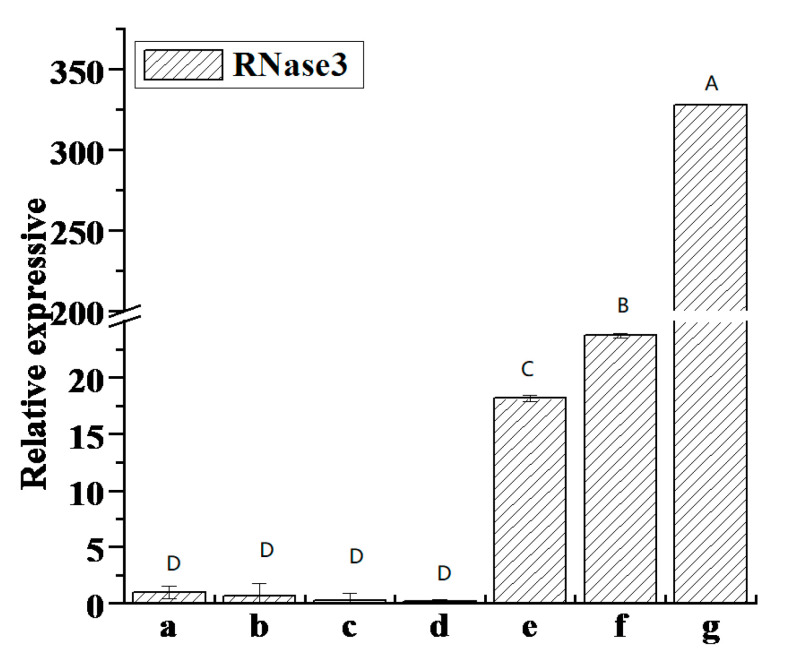
Relative expression levels of the *RNase3* gene in sweet potato samples. (a–g) Jin 3–82 (green leaf), Jin 3–82 (yellow leaf), Guang 87–72 (July), Guang 87–72 (April), Mianshu 8, Jin 3 (July), and Jin 3 (May). Error bars represent the standard error (*n* = 3) for each group. The capital letters in the figure indicate significant differences.

**Table 1 cimb-47-00242-t001:** Prevalence and types of sweet potato virus diseases in different regions of Fujian Province.

Virus Species	Fuzhou Area		Quanzhou Area		Putian Area		Longyan Area		Sanming Area		Total	
	Infection Number	Detection Rate	Infection Number	Detection Rate	Infection Number	Detection Rate	Infection Number	Detection Rate	Infection Number	Detection Rate	Infection Number	Detection Rate
SPFMV	26	60	10	91	2	50	0	0	0	0	38	60
SPCSV	11	26	4	36	2	50	0	0	0	0	16	25
SPVC	13	30	3	27	3	75	0	0	0	0	19	30
SPVG	3	7	3	27	1	25	0	0	0	0	7	11
SPV2	19	44	2	18	4	100	2	100	1	33	28	44
SPLV	1	2	1	9	0	0	0	0	0	0	2	3
SPVMV	0	0	2	18	0	0	0	0	0	0	2	3
SPCFV	0	0	0	0	0	0	0	0	0	0	0	0
SPMMV	0	0	0	0	0	0	0	0	0	0	0	0
CMV	0	0	1	9	0	0	0	0	0	0	1	2
SPLCV	7	16	0	0	1	25	0	0	0	0	8	16
SPPV	20	47	4	36	1	25	2	100	2	67	29	58
SPSMV-1	21	49	5	45	0	0	2	100	3	100	31	62
TLCV	0	0	0	0	0	0	0	0	0	0	0	0

**Table 2 cimb-47-00242-t002:** Detection results of sweet potato seedling virus diseases across different growth stages.

Virus Species	Period			
	Sweet Potato Seedling Stage	Pre Growth Stage	Peak of Growth	Anaphase
SPFMV	79	63	44	100
SPCSV	43	50	0	15
SPVC	36	44	11	23
SPVG	21	6	11	8
SPV2	64	88	56	0
SPLV	14	0	0	0
SPVMV	7	0	11	0
CMV	7	0	0	0
SPLCV	14	14	38	15
SPPV	64	57	50	70
SPSMV-1	71	43	75	70

**Table 3 cimb-47-00242-t003:** SPVD detection rates at different growth stages of sweet potatoes.

Virus Species	Period			
	Seedling Stage	Pre Growth Stage	Peak of Growth	Anaphase
SPFMV	79	63	44	100
SPCSV	43	50	0	15
SPVD	27	64	0	15

**Table 4 cimb-47-00242-t004:** Occurrence of sweet potato virus diseases in different varieties in Fujian Province.

Variety Name	Virus Name										
	SPFMV	SPCSV	SPVC	SPVG	SPV2	SPLV	SPVMV	CMV	SPLCV	SPPV	SPSMV-1
LY1					+					+	+
LY2					+					+	+
SM1					+					+	+
SM2										+	+
SM3											+
Shenglibaihao										+	
Jin763											
Jin11											+
Jin208											
FZ1										+	+
Rong910										+	
11-500										+	
QZJJC1					+		+			+	+
QZJJC4	+		+	+							
FZ2	+									+	+
FZ3	+		+							+	+
Guang08-6	+		+							+	+
Zhan271	+		+						+	+	+
Zhan11	+				+				+	+	+
Guagn79	+		+	+					+	+	
Fu18	+	+	+								
Quan32	+	+		+						+	+
Puhang1hao	+									+	+
Quan12	+	+		+				+		+	+
13286	+	+	+							+	+
Mianshu8hao	+	+	+		+	+	+				
Jin2-1	+				+					+	+
Guang79-44	+										+
Guang87-57	+				+				+	+	+
Jin3	+	+	+		+					+	
Guagn87-72	+	+	+		+						+
Jin57	+		+	+	+	+					+
FZ4					+					+	+
Pu3hong					+						
Jin15		+	+		+				+	+	
Longjinshu1hao	+	+	+	+	+						
Funingzi3hao					+						+
Long710	+				+				+	+	+
Puzi5hao										+	+
Jin3-97					+						+
Jin3-82	+	+	+		+				+		+

Note: “+” indicates virus detected; blank indicates that no virus is detected.

**Table 5 cimb-47-00242-t005:** Testing results of SPVD in different regions in Fujian Province.

Virus Type	Fuzhou		Quanzhou		Putian		Longyan		Sanming		Total	
	Infection Number	Detection Rate	Infection Number	Detection Rate	Infection Number	Detection Rate	Infection Number	Detection Rate	Infection Number	Detection Rate	Infection Number	Detection Rate
SPFMV	26	60	10	91	2	50	0	0	0	0	38	60
SPCSV	11	26	4	36	2	50	0	0	0	0	16	25
SPVD	7	21	3	27	2	50	0	0	0	0	12	19

**Table 6 cimb-47-00242-t006:** Occurrence of sweet potato virus diseases in different parts in Fujian Province.

Virus Species	Leaf				Tuber			
	Guang79	Fu18	Zhan271	Quan12	Guang79	Fu18	Zhan271	Quan12
SPFMV	+	+	+	+	+	+	+	+
SPCSV				+		+		
SPVC	+				+	+	+	
SPVG	+			+				+
SPV2								
SPLV								
SPMMV								
CMV				+				
SPLCV	+		+					
SPPV	+		+					+
SPSMV-1			+					+

## Data Availability

Data are contained within the article and Appendix A.

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
