# Peer review of "Key Sweet Potato Viruses in Fujian Province and Their Distribution, Harmfulness, and Implications in China"

_cimb, 2025, doi:10.3390/cimb47040242_

Round 1
Reviewer 1 Report
Comments and Suggestions for Authors
Please see the attached file.

Author Response
Title:
- The word "Harmfullness" should be corrected to "Harmfulness."
Response: Thank you very much, we have changed "Harmfullness" to "Harmfulness.
Abstract:
- The phrase "China, the world’s largest sweet potato producer, faces significant
threats to sweet potato production from virus diseases" could be rewritten for
conciseness:
· "China, the largest global producer of sweet potatoes, faces significant
threats from viral diseases, particularly in Fujian Province, where sweet
potatoes are the second most important food crop after rice."
Response: Thank you very much, we have changed the sentence to "China, the largest global producer of sweet potatoes, faces significant threats from viral diseases, particularly in Fujian Province, where sweet potatoes are the second most important food crop after rice."
- In “Damage was most severe during the seedling stages and lighter during the
growth period”, specify whether this was observed for all viruses or primarily for
SPVD.
Response: Thank you for your question. We have revised the sentence as “Especially for SPVD, damage was most severe during the seedling stages and lighter during the growth period”.
- The sentence "These findings provide a scientific reference for managing sweet
potato virus diseases." could be more specific:
· "These findings offer insights into the epidemiology of sweet potato viruses
and serve as a reference for developing targeted disease management
strategies."
Response: Thank you very much, we have changed the sentence to“These findings offer insights into the epidemiology of sweet potato viruses and serve as a reference for developing targeted disease management strategies.”
Introduction:
1. There are minor grammatical errors in several places. For instance:
· “Sweet potato was introduced to China from Fujian Province and plays a vital
role in food security, serving as the second most produced food crop after rice in Fujian.”
- Consider revising to: “Sweet potato, introduced to China through Fujian Province, plays a crucial role in food security and is the second most cultivated food crop in Fujian after rice.”
Response: Thank you very much, we have changed the sentence to “Sweet potato, introduced to China through Fujian Province, plays a crucial role in food security and is the second most cultivated food crop in Fujian after rice.” And checked the whole paragraph to corrected some grammatical errors.
- The sentence “More than 30 virus species, spanning families such as Bromoviridae,
Bunyaviridae, Cauimoviridae, Closteroviridae, etc.”
- The use of "etc." is discouraged in scientific writing. Listing key virus families
suffices.
Response: Thank you very much, we have deleted “etc.”
- The manuscript mentions that SPCSV’s RNase3 gene plays a key role in SPVD severity. However, more details on its mechanism of action would strengthen the discussion.
Response: Thank you for your suggestion, we have added details on its mechanism of action in the discussion, please see in the new revise article.
Materials and methods:
· The phrase “Nucleic acid was achieved using the steel ball method for leaves and mortar method for roots” is unclear and should be revised for clarity.
Response: Thank you,the sentence was revised as“The sample of leaves were grinded adding sterilized small steel ball in the tubes, and the roots were grinded in the sterilized mortars.” please see in the new revise article.
Experimental Details:
- The rationale for selecting the 63 samples (41 varieties) should be better justified.
Response: Thank you for your suggestion, these selected sampling locations are important sweet potato production area in Fujian Province, and the sampled varieties are mostly planted in Fujian Province. We have revised the phrase in the revised manuscript.
- The criteria for primer selection should be elaborated further. Were these
primers validated in previous studies? Were they tested for cross-reactivity?
Response: Thank you, these primers were selected according to literature, and were tested and validated in previous. We have revised the phrase in the manuscript.
- The qRT-PCR section should clarify how the reference gene (GAP) was selected and validated.
Response: Thank you, the reference gene (GAP) was selected according to literature (Park S C, Kim Y H, Ji C Y, et al. Stable internal reference genes for the normalization of real-time PCR in different sweet potato cultivars subjected to abiotic stress conditions. PloS one, 2012, 7(12): e51502). Actually, we selected one from two primers and validated previously. We have revised the phrase in the manuscript.
- Were biological replicates included? The manuscript states three technical replicates, but biological replicates are critical for validation.
Response: Thank you, the samples were collected based on the symptoms of the leaves during the investigation on the field, in this study, only technical replicates were used.
Minor Suggestions
· The study should define all abbreviations at first use (e.g., EB, GAP).
Response: Thank you for your suggestions, we have defined the abbreviations at first use.
- The materials and methods section should be written in past tense for consistency.
Response: Thank you for your suggestions, we have checked and revised the materials and methods section in past tense.
- Consider reformatting long paragraphs for better readability.
Response: Thank you for your suggestions, we have reformatted into short paragraphs in the manuscript.
Conclusion:
1. The phrase "Sweet potato virus diseases, especially SPVD virus disease, are generally showing a trend of gradually expanding damage, and effective measures are urgently needed to prevent and control them."
- Consider simplifying: "Sweet potato virus diseases, particularly SPVD, are increasingly spreading, highlighting the urgent need for effective prevention and control strategies."
Response: Thank you very much, we have changed the sentence to "Sweet potato virus diseases, particularly SPVD, are increasingly spreading, highlighting the urgent need for effective prevention and control strategies." please see in the new revise article.
- The statement “Further research is needed to determine whether altering RNase3 expression can mitigate SPVD symptoms or inhibit its occurrence” is important. However, a suggestion on potential experimental approaches (e.g., gene silencing or CRISPR-based approaches) would strengthen the future research outlook.
Response: Thank you, we have revised the sentence in the manuscript base on your suggestion, please see in the new revise article.
Final Recommendation:
Minor Revisions Required.
Some linguistic and structural refinements need implementation to achieve scientific precision along with clarity in the well-designed study. Better development of discussion concerning RNase3 in combination with conclusion elaboration and proper grammar will enhance the manuscript quality
Response: Thank you very much for your recommendation, we have revised linguistic and structural, Supplemented the discussion on RNase3 gene. please see in the new revise article.
Reviewer 2 Report
Comments and Suggestions for Authors
In this manuscript, Weikun Zou and Xuanyang Chen identified 11 viruses, including SPFMV and SPCSV, infecting sweet potatoes in Fujian, with sequence comparisons revealing diverse strains from many sources. I have following comments:
1, For the title, I suggest to employ “Key Sweet Potato Viruses Identified from Fuijan Province and their Distribution, Harmfullness, and Implications in China”.
2, For the Abstract, full names of SPFMV, SPCSV, SPV2, SPSMV-1, SPPV, and SPVD should be provided. More values and details should be provided. For instance, authors stated that “Damage was most severe during the seedling stages and lighter during the growth period.”, please provide the value.
3, For the key words, “Fuijan Province” should be included.
4, For the introduction, main conclusion and the practical interest of this study should be stated in the last paragraph.
5, For the results, Table 2 is missing!!! scale bars should be included in Figure 2, and error value should be provided in Tables 1, 3, 4, and 6. Results of significance difference analysis should be presented in Figure 4.
6, For the materials and methods, genotypes of Sweet Potato Viruses examined in this study should be described. Biological and technical replicates, as well as randomization methods should be clearly stated. Sampling size should be included.
7, For the discussion, I would like to see the discussion section was divided into subsections with appropriate titles.
Author Response
1, For the title, I suggest to employ “Key Sweet Potato Viruses Identified from Fuijan Province and their Distribution, Harmfullness, and Implications in China”.
Response: Response: Thank you very much, we have changed the title to “Key Sweet Potato Viruses Identified from Fuijan Province and their Distribution, Harmfullness, and Implications in China”.
2, For the Abstract, full names of SPFMV, SPCSV, SPV2, SPSMV-1, SPPV, and SPVD should be provided. More values and details should be provided. For instance, authors stated that “Damage was most severe during the seedling stages and lighter during the growth period.”, please provide the value.
Response: Thank you for your suggestion, the full names of SPFMV, SPCSV, SPV2, SPSMV-1, SPPV, and SPVD were provided, More values and details were provided. please see in the new revise article.
3, For the key words, “Fuijan Province” should be included.
Response: Thank you for your suggestion, we have added the key words “Fuijan Province”.
4, For the introduction, main conclusion and the practical interest of this study should be stated in the last paragraph.
Response: Thank you for your suggestion, we added main conclusion and the practical interest in the last paragraph. please see in the new revise article.
5, For the results, Table 2 is missing!!! scale bars should be included in Figure 2, and error value should be provided in Tables 1, 3, 4, and 6. Results of significance difference analysis should be presented in Figure 4.
Response: we are sorry for the error of Table 2, now we have revised the table number. We add the scale bar in Fig 2. The data in Table 1,3,4,6 are from samples of the sweet potato leaves with symptoms of viral disease during survey in the production fields, without duplication and statistical analysis, so there is no error value. The significance difference analysis was finished in the Fig.4.
6, For the materials and methods, genotypes of Sweet Potato Viruses examined in this study should be described. Biological and technical replicates, as well as randomization methods should be clearly stated. Sampling size should be included.
Response: Thank you for your suggestion, we have described some varieties names in the paragraph, and listed all the varieties in table 5. In this study, the samples were collected based on the symptoms of the leaves during the investigation on the field, in this study, only technical replicates were used. One or more leaves or tuberous roots were collected according to the symptom of virus disease.
7, For the discussion, I would like to see the discussion section was divided into subsections with appropriate titles.
Response: Thank you for your suggestion, we have divided the discussion into section with appropriate titles.
Round 2
Reviewer 2 Report
Comments and Suggestions for Authors
The revised manuscript has been greatly improved.